# Dynamically Tunable Resonant Strength in Electromagnetically Induced Transparency (EIT) Analogue by Hybrid Metal-Graphene Metamaterials

**DOI:** 10.3390/nano9020171

**Published:** 2019-01-30

**Authors:** Chaode Lao, Yaoyao Liang, Xianjun Wang, Haihua Fan, Faqiang Wang, Hongyun Meng, Jianping Guo, Hongzhan Liu, Zhongchao Wei

**Affiliations:** Guangdong Provincial Key Laboratory of Nanophotonic Functional Materials and Devices, School of Information and Optoelectronic Science and Engineering, South China Normal University, Guangzhou 510006, China; laochaode@m.scnu.edu.cn (C.L.); lyy777@m.scnu.edu.cn (Y.L.); 2016021810@m.scnu.edu.cn (X.W.); fqwang@scnu.edu.cn (F.W.); hymeng@scnu.edu.cn (H.M.); guojp@scnu.edu.cn (J.G.); lhzscnu@163.com (H.L.)

**Keywords:** EIT analogue, tunable EIT, graphene ribbon, resonance intensity

## Abstract

In this paper, a novel method to realize a dynamically tunable analogue of EIT for the resonance strength rather than the resonance frequency is proposed in the terahertz spectrum. The introduced method is composed of a metal EIT-like structure, in which a distinct EIT phenomenon resulting from the near field coupling between bright and dark mode resonators can be obtained, as well as an integrated monolayer graphene ribbon under the dark mode resonator that can continuously adjust the resonance strength of transparency peak by changing the Fermi level of the graphene. Comparing structures that need to be modulated individually for each unit cell of the metamaterials, the proposed modulation mechanism was convenient for achieving synchronous operations for all unit cells. This work demonstrates a new platform of modulating the EIT analogue and paves the way to design terahertz functional devices which meet the needs of optical networks and terahertz communications.

## 1. Introduction

EIT is a concept describing a sharp transmission window generated in a broad absorption profile, and was first discovered in the process of atomic system experiments [1,2,3]. Due to the transparent peak of the transmission window having a great influence on the dispersion parameters of the transmitted light, the EIT possesses many potential applications for slow light control, bio-chemical sensing, filtering, absorbing, switching and so on [4,5,6,7,8]. Conventional generation of EIT phenomena in an atomic system experiment requires extremely low temperatures and very stable optical pumping sources, which hindered the development of EIT devices in practical applications. In recent years, the analogues of EIT in metamaterials have been the most intensively investigated analogues because of their favorable properties such as flexible design and easy realization. However, most EIT-analog metamaterials are composed of common noble metal materials with a fixed response frequency under fixed structural parameters where the dynamically adjustable function is usually realized by changing the structural parameters of the metamaterial, which is very inconvenient and impractical. Even though some methods have been put forward to dynamically modulate the analogues of EIT in metamaterials, such as changing the incident light angles [9,10] or tuning an external magnetic field [11], their modulation methods remain complicated and the modulation response speeds (MRS) are still very low.

More recently, graphene, composed of carbon atoms and sp2 hybrid orbital hexagonal honeycomb crystal lattice, have attracted a lot of attention due to their high MRS on the order of picoseconds, adjustable conductivity and other excellent optical, electrical and magnetic properties compared to conventional noble metal materials [12,13]. In view of these exceptional properties, a number of graphene-based metamaterial devices with excellent function have been put forward, such as terahertz absorber, beam manipulating device and analog computing array [14,15,16]. Besides, it can be used as an active material [17,18,19] embedded in metamaterials and is a superior candidate for dynamic adjustment of the EIT analogues. However, most of the research focusses on adjusting the positions of resonant frequencies instead of the resonance strength of the EIT analogues. Besides, the modulators based on large area graphene have a relatively low modulation speed. Although the modulation speed can be increased by using a discrete structure, the complexity of the structure makes it difficult to apply in practice. Thus, new research about the diversity and functionality of controllable EIT analogue effects with higher modulation speed and more accessible fabrication tech are in particularly high demand due to their widespread applications in optical networks and terahertz communications.

In this paper, an innovative method to realize dynamically tunable EIT for the resonance strength rather than the resonance frequency of the transmission windows is proposed at the terahertz spectrum. Apart from the demonstration of the EIT analogue phenomenon by an all-new structure, in which the dark and bright mode resonator are represented by a metal split ring and a cut wire respectively, we integrated monolayer graphene ribbons under the dark mode resonator and then realized the complete modulation for the resonant intensity of the EIT-like transmission window by manipulating the Fermi level of the graphene. It is notable that in the manipulation process there is no need to change the structure parameters and the position of resonant frequency remains unchangeable. Comparing previous structures with discrete modulations for each unit cell, the proposed modulation mechanism is convenient and fast for it can achieve synchronous operations for all unit cells of the metamaterials, which makes the design more accessible for practical applications. In addition, the corresponding group index changed during the modulation process, which proves that the method can obtain a better controllable slow light effect. This work contributes a strategy to increase the diversity and functionality of controllable EIT analogue effects and opens a new platform of modulating the EIT analogue with higher modulation speed and easier fabrication tech for designing novel terahertz functional devices.

## 2. Materials and Methods

Figure 1a introduces the schematic diagram of the tunable metamaterial device: a hybrid metal-graphene terahertz metamaterial is designed to actively control EIT. The periodically structure is deposited on the substrate. The Fermi level of the graphene ribbon connected to the electrode can be changed by changing the gate voltage between the substrate and the electrode at the edge of the metamaterial. The metal electrode is made of gold (Au), and the substrate is made of dielectric with a relative permittivity of 11.7. Figure 1b depicts the detailed structural parameters of a single substructure of metamaterial. The unit cell of the metamaterial can be decomposed into two constituent elements: a split ring (SR) resonator and a cut wire (CW) resonator. CW is used as a bright mode resonator and SR is used as a dark mode resonator. The transmission spectra of three metamaterial components including an SR, a CW, and the entire structure as shown in Figure 1c,d respectively. As shown in Figure 1c, the bright mode resonator has a broad absorption peak in the terahertz band, and its resonance peak is at a specific position of 1.6 THz. The transmission spectrum of the proposed structure is the result of the analog of EIT effect under incident plane wave illumination in the x-polarization direction as shown in Figure 1d. There is a sharp transparent peak at the center of the broad absorption peak, and its position coincides with the position of the absorption peak of the bright mode resonator as 1.6 THz. The material used for the bright and dark mode resonator is aluminum (Al) metal. The graphene ribbon which is placed under the left panel of SR serves as the “wire” connecting all SRs of the metamaterial.

The thickness of Al and substrate are 200 nm and 400 nm, respectively. The optical properties of metallic Al can be expressed using the Drude model [20]:(1)$$\varepsilon_{Al}=\varepsilon_{\infty }-\frac{\omega_{P}^{2}}{\omega^{2}+i\omega \gamma }$$

In the formula, γ is the damping coefficient and ω_p_ is the plasma frequency, and the values are 1.22 × 10^14^ rad/s and 2.24 × 10^16^ rad/s respectively.

For the graphene material to be added, its conductivity is a major factor affecting its properties. According to the Kubo equation [21], the conductivity of graphene consists of the in-band conductivity and the inter-band conductivity [22]:(2)$$\begin{array}{c} \sigma_{g}=\sigma_{\mathrm{in}}+\sigma_{\mathrm{int}er}\\ =\frac{2e^{2}K_{B}T}{\pi \hbar^{2}}\frac{i}{\omega +i\Gamma^{-1}}\ln\left[2\cosh\left(\frac{E_{F}}{2K_{B}T}\right)\right]+\\ \frac{e^{2}}{4\hbar }\left[\frac{1}{2}+\frac{1}{\pi }\tan^{-1}\left(\frac{\hbar \omega -2E_{F}}{2K_{B}T}\right)-\frac{i}{2\pi }\ln\frac{{\left(\hbar \omega +2E_{F}\right)}^{2}}{{\left(\hbar \omega -2E_{F}\right)}^{2}+4{\left(K_{B}T\right)}^{2}}\right] \end{array}$$
where e is the electron charge, k_B_ is the Boltzmann constant, T is the temperature of the environment (T = 300 K), ħ is the simplified Planck constant, and ω is the incident plane wave angle frequency. In particular, the Fermi level of graphene is represented by E_F_ in the formula, and the average relaxation time is represented by Γ.

In the terahertz band, the in-band partial conductivity of graphene is negligible for the electrical conductivity of the total graphene. When graphene is highly doped, condition E_F_ ≫ K_B_T and E_F_ ≫ ћω is satisfied to obtain a simplified Drude formula [23].
(3)$$\sigma_{g}={\left(\frac{e}{\hbar }\right)}^{2}\frac{E_{F}}{\pi }\frac{i}{\omega +i\Gamma^{-1}}$$

The carrier relaxation time define as Γ = (μE_F_)/(e$v_{F}^{2}$), where v_F_ is the Fermi velocity and set as 1.1 × 10^6^ m/s. μ is the carrier mobility, and the value is the experimentally measured stable value 3000 cm^2^/(V·s) [24].

Then, as shown in Figure 2, the conductivity of graphene is different under different Fermi levels of graphene. The graphene Fermi level can be changed by applying electric gate voltage between the substrate and the electrode at the edge of the metamaterials. It can be seen from the figure that in the terahertz band selected in this paper, the real and imaginary parts of the cpplying electric gate voltage between the substrate and the electrode at the edge of the metamaterialonductivity of graphene under different Fermi levels have different values.

## 3. Simulation Results

These simulation experiments used FDTD Solution. Precious metal structures at sub-wavelength scale can excite local surface plasmons (LSP) under certain conditions. The electric field distribution of the bright mode structure shown by Figure 3a,d shows that the proposed bright mode resonator can excite a partial surface plasmon under the incident plane wave irradiation in the y polarization direction. The bright mode resonator usually exhibits strong coupling with incident plane waves. Figure 3b,e are electric field distribution diagrams of the dark mode resonator showing that the dark mode structure has a very weak electric field distribution and cannot directly couple to the external field. But it can be excited by the local field caused by the bright mode via near field coupling. The destructive interference between distributions at the resonance frequency verified an electric dipole mode and confirmed that the CW bright and dark mode resonator leads to an outstanding transparency window at the bright mode resonant frequency. The electric field can be used as a bright element. The Figure 3c,f shows the electric field layout of the metamaterial at 1.6 THz. Strong electric fields concentrated around two ends of SR demonstrated that the SR turns into monopole mode after exciting by the CW. It is the destructive interference between the two different modes of monopole mode and dipole mode that leads to the generation of EIT analogue effects.

Subsequently, in order to conveniently obtain and feasibly control the transmission window of the EIT analogue, the controllable material graphene is added between the dark mode resonator and the substrate. In addition, the graphene ribbon can be connected to all subunit structures on the metamaterial to allow the device to be adjusted simultaneously. Therefore, as displayed in Figure 1a,b, this is a simple and workable method to achieve the electrically controllable EIT analogue by adding bias voltage to graphene. In particular, the spectral line can be altered without rebuilding the structure or changing the position of the frequency response in the process of regulating the Fermi level of graphene. As shown in Figure 4a–d, as the Fermi level of graphene increased, the height of the transmission windows is gradually reduced. Therefore, the resonant intensity is changed instead of the resonant frequency during the modulation process. At first, in the absence of graphene, there is no detuning between the SR and CW resonator, and the position of the transmission peak is 1.6 THz. After adding graphene with a Fermi level of 0.2 eV, there is a significant decline of the transmission peak. As the Fermi level E_F_ in the graphene ribbon increased from 0.2 eV to 0.8 eV, the summit of the spectrum starts to disappear gradually. The transmission spectrum changes from a symmetrical Lorentzian line shape to an asymmetrical Fano line shape. Consequently, complete modulation of the EIT analogue is obtained.

## 4. Discussion

Since graphene is only added to the dark mode resonator, a dark mode resonator is needed to cause a change in the EIT-like transmission spectrum. Next, the coupled harmonic oscillator model will be combined, and then the electric field distribution of the structure during the adjustment process will be analyzed to analyze which properties of the dark mode resonator specifically affect by the graphene [25]. The coupled differential equations of the two resonators are as follows.
(4)$${\ddot{\chi }}_{b}+\gamma_{b}{\dot{\chi }}_{b}+\omega_{0}^{2}\chi_{b}-k\chi_{d}=E$$
(5)$${\ddot{\chi }}_{d}+\gamma_{d}{\dot{\chi }}_{d}+{\left(\omega_{0}+\delta \right)}^{2}\chi_{d}-k\chi_{b}=0$$

The resonant amplitude and damping coefficient of the bright mode resonator are χ_b_ and γ_b_, respectively, while the resonant amplitude and damping coefficient of the dark mode harmonic oscillator are χ_d_ and γ_d_. The resonant frequency of bright mode resonator under incident plane wave illumination is denoted by ω_0_, and its value is 2π × 1.6 THz. δ is the detuning frequency between the two oscillators, and k is the coupling coefficient between the monopole mode and the dipole mode. By way of solving the Equations (4) and (5), the susceptibility χ can be gained by γ_b_, γ_d_, δ and k with the approximate formula ω−ω_0_ ≪ ω_0_ [26,27].
(6)$$\chi =\chi_{r}+i\chi_{i}\propto \frac{\left(\omega -\omega_{0}-\delta \right)+i\frac{\gamma_{d}}{2}}{\left(\omega -\omega_{0}+i\frac{\gamma_{b}}{2}\right)\left(\omega -\omega_{0}-\delta +i\frac{\gamma_{b}}{2}\right)-\frac{k^{2}}{4}}$$

The loss of energy in the metamaterial device is mainly determined by the imaginary part of the susceptibility. The transmittance of a metamaterial device under plane wave illumination can be expressed by the following Equation (7).
(7)$$T=1-s\chi_{i}$$

In the above formula, the intensity coefficient of the coupling between the incident plane wave and the mode harmonic oscillator is denoted by s. The red dashed line in Figure 4a–d represents the derivation and fitting results based on the coupled harmonic oscillator model.

Figure 5b shows the fitted values of the correlation coefficients in the coupled harmonic oscillator model for adjusting the graphene Fermi level. As the picture shows, γ_b_, δ and k hardly changed and γ_d_ gradually increased during the process of increasing E_F_. Among them, the values of k, γ_b_ and δ were maintained at around 150 rad/ps, 2.5 rad/ps and 0 rad/ps before and after the addition of graphene, respectively. The value of γ_d_ is gradually increased from 0.5 rad/ps at the beginning to 5 rad/ps at the end. γ_b_ remains unchanged, indicating that graphene does not affect the damping coefficient of the bright mode resonator, and cannot change the inherent energy loss of the bright mode resonator. The constant value of K means that there is no change in the coupling between the dipole mode and the monopole mode. The value of δ close to zero remains unchanged, which indicates that there is no detuning between the bright mode resonator and the dark mode resonator. The damping coefficient of the dark mode resonator has a large variation in the process of adjusting the graphene Fermi level, which is the main factor affecting the transmission spectrum of the metamaterial. When the graphene ribbon is placed under a dark mode resonator and the Fermi level of graphene is gradually increased, the loss of the dark mode resonator is also increased. When the damping coefficient of the dark mode resonator is too large, the dark mode resonator will not maintain the resonance state due to excessive loss. Therefore, when the Fermi level modulation of graphene is 0.8 eV, the analogue of EIT phenomenon on the transmission spectrum of the metamaterial substantially disappears.

Combined with the electric field distribution diagram, the electric field distributions at the transmission windows of EIT analogue during the process of modulating the Fermi level of graphene is plotted. As shown in Figure 4e, there is a traditional electric field distribution of the analogue of EIT phenomenon caused by destructive interference between bright and dark mode resonators. Strong electric fields are distributed at both ends of the dark mode structure, so the damping coefficient of the dark mode structure is low. Figure 4f–h shows the electric field distribution of the metamaterial at 1.6 THz after adding a graphene ribbon to the dark mode resonator and changing the Fermi level of graphene from 0.2 eV to 0.8 eV. As shown, as the graphene Fermi level increases, the electric field of the dark mode structure becomes weaker. Therefore, the losses of the dark mode structure continued to increase with the increasing of the Fermi level of graphene, which agree very well with the results suggested by the theory. In the proposed hybrid metal-graphene structure, there is high conductivity of the graphene, which exist the properties of quasi-metal and enhance the losses of dark mode resonator to decline the destructive interference. When the loss of the dark mode resonator caused by graphene is too large, the interference phenomenon between the dark mode resonator and the bright mode resonator disappears, so that the analogy of EIT phenomenon disappears.

In the transparency window of EIT analogue, there is strong dispersion, which means it can slow down the speed of light [28]. The group index n_g_ is employed to represent the capability of slow light, which can be expressed as [29]:(8)$$n_{g}=n_{e}+\omega \frac{dn_{e}}{d\omega }$$

By extracting the S parameters, the effective refractive index n_e_ of the metamaterial can be obtained [30]. Figure 6a–d shown the group index and the imaginary part of the refractive index of the proposed hybrid metal-graphene metamaterials. Figure 5a shows the distribution of the maximum value of the imaginary part of the effective refractive index and the group index. By comparison of the maximum values, a larger group index and a smaller imaginary part of the effective refractive index can be obtained at the corresponding transparent window. That is, the metamaterial has lower loss while achieving better slow light performance. In the process of changing the Fermi level of graphene, the group index gradually decreases. In addition, during the adjustment of the graphene Fermi level, the group index varies widely. This shows that the slow light performance of the metamaterial device proposed in this paper has a wide controllable range.

## 5. Conclusions

In summary, we have proposed and demonstrated a method to realize a dynamically tunable EIT phenomenon for the resonance strength instead of the resonance frequency of the transmission windows. The transmission spectrum of the metamaterial shows a pronounced transmission peak at 1.6 THz, which is caused by the destructive interference resulting from near field coupling of two resonance modes. Therefore, the complete modulation of the EIT analogue at the specific frequency can be realized without affecting adjacent spectrum by changing the Fermi level of graphene. Furthermore, the coupled harmonic oscillator model is adopted to describe the near field coupling effect and analyze the physical mechanism by the fitting parameter. In the meanwhile, the control of the group index is also calculated for the applications of slowing light. This work offers a new perspective on the design of dynamically adjustable EIT-like devices. Moreover, by changing the size of the device, the design scheme can be easily scalable to other terahertz, infrared or visible regimes for various promising applications in optical networks and terahertz communications.

## Figures and Tables

**Figure 1 nanomaterials-09-00171-f001:**
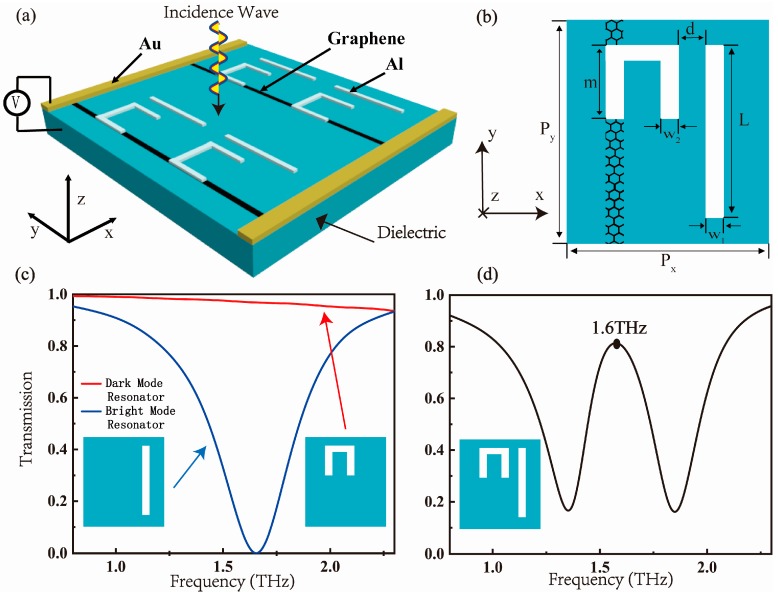
(**a**) The schematic representation of a metamaterial device under a plane wave incident; (**b**) the geometric parameters of a single substructure of metamaterials: P_x_ = 80 µm, P_y_ = 120 µm, L = 80 µm, m = 28 µm, w_1_ = 5 µm, w_2_ = 4 µm, d = 3 µm perpendicularly. (**c**) shows the transmission spectrum of the bright and dark modes resonator, and (**d**) shows the transmission spectrum of the EIT analogue structure combined with the bright and dark modes resonator under the incident plane wave with polarization direction y.

**Figure 2 nanomaterials-09-00171-f002:**
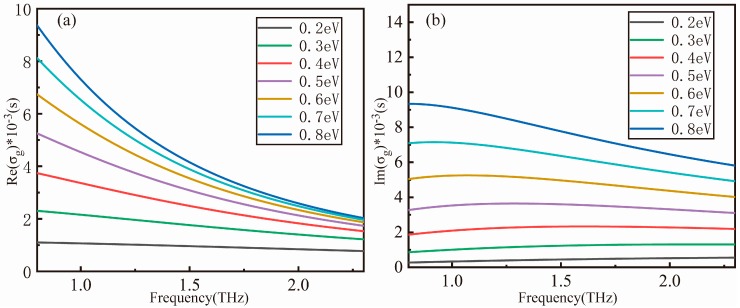
The conductivity of graphene calculated according to the formula is divided into (**a**) real and (**b**) imaginary parts.

**Figure 3 nanomaterials-09-00171-f003:**
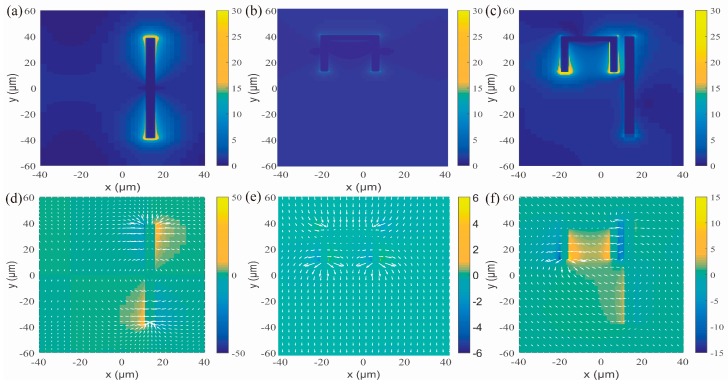
x-component of electric field and electric field vector size and direction of (**a**) the bright mode resonator (**b**) the dark mode resonator and (**c**) the combined EIT analogue structure. The magnitude of electric field of (**d**) the bright mode resonator (**e**) the dark mode resonator and (**f**) the EIT analogue structure. All of the above are the results of illumination of an incident plane wave at a frequency of 1.6 THz.

**Figure 4 nanomaterials-09-00171-f004:**
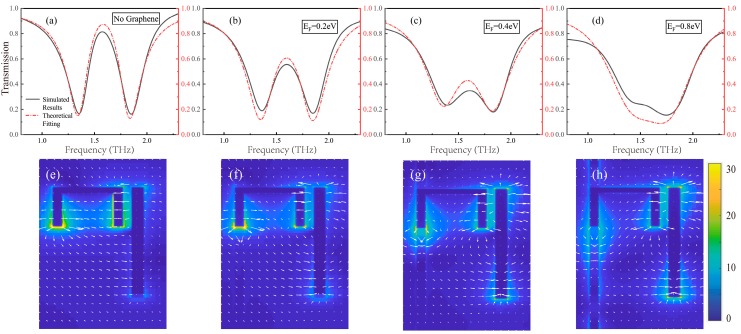
(**a**–**d**) Transmission spectrum of the proposed metamaterial structure at different Fermi levels of graphene and the corresponding theoretical fitting results. Correspondingly, the electric field distribution of the metamaterial substructure unit at a frequency of 1.6 THz when there is no graphene (**e**) and the graphene Fermi level is 0.2 eV (**f**) 0.4 eV (**g**) and 0.8 eV (**h**) respectively.

**Figure 5 nanomaterials-09-00171-f005:**
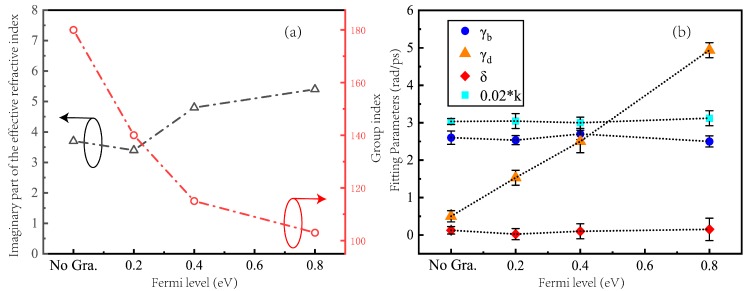
(**a**) The maximum distribution of group index and imaginary part of the effective refractive index when no graphene and graphene Fermi levels are 0.2 eV, 0.4 eV and 0.8 eV. (**b**) The fitted values of the correlation coefficients in the coupled harmonic oscillator model with the Fermi level of graphene varying from 0.2 eV to 0.8 eV. The unit of k is rad^2^ ps^−2^.

**Figure 6 nanomaterials-09-00171-f006:**
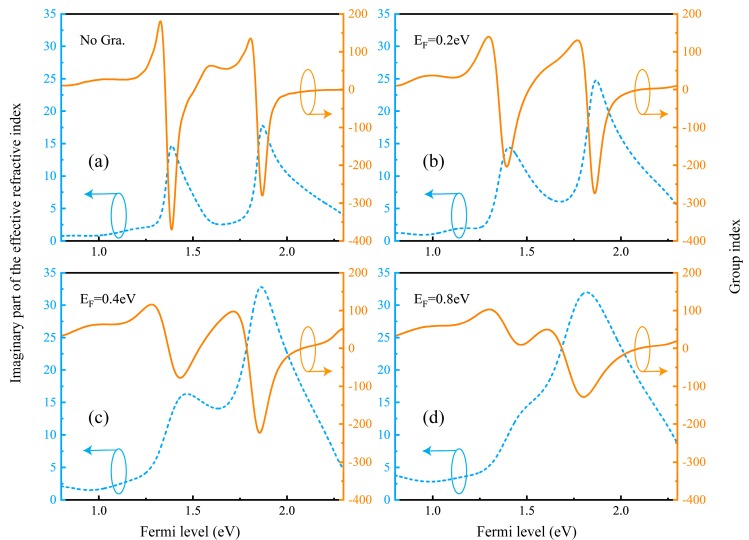
(**a**–**d**) The group index n_g_ and the imaginary part of the effective refractive index Im (n_e_) for the proposed hybrid metal-graphene metamaterials with various Fermi level E_F_ of graphene.

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
