# Peer review of "Dynamically Tunable Resonant Strength in Electromagnetically Induced Transparency (EIT) Analogue by Hybrid Metal-Graphene Metamaterials"

_nanomaterials, 2019, doi:10.3390/nano9020171_

Round 1
Reviewer 1 Report
This manuscript described the dynamically tunable EIT structure with graphene hybrid structure.
Could you verify the position of peak point during changing the Fermi level of graphene layer in Fig. 4?
For practical application, could you estimate the delay of switching speed due to the parasitic capacitance resulted from the overlap of metal layer with substrate (It can be related with the size (or number) of array)?
Author Response
A detailed response to the reviewer's comments is in the uploaded document.

Reviewer 2 Report
The author present a EIT analogue system, consisting of a dark and bright mode resonator. A strip of graphene is used to tune the strength of the EIT-like response.
The proposed work is scientifically sound, but there are some issues with the manuscript that need addressing prior to publication.
1. Several of the references are incorrect. For example, references 9-11 are given as applications of EIT like responses. But none of these three is concerned with an EIT like response.
2. In figure 4 panels 1-d. There is at times a significant difference between the theoretically predicted and the simulated response. What is the origin of this discrepancy? Is it an error from the simulations, e.g. a too large mesh? Or is it from assumption in the theoretical analysis, e.g. the Drude model not being a proper model of materials involved? Since the discrepancy is on the same order as the one expected between a theoretical and experimental result it needs to be discussed appropriately.
3. The authors repeatedly state that “a strip of graphene between the resonators” is used. This is misleading, as the graphene is not between the resonators. It is near the far end of the dark mode resonator.
4. What is the impact of changing the fermi level of the graphene on the resonance spectrum of the dark mode resonator. This should be discussed with figures.
5. How are the transmission spectra in fig.1 obtained? What is the resonance spectrum (not transmission spectrum) of the dark mode resonator?
6. The authors do include a discussion of the underlying physics, but this should already be referred to results are presented. Currently the paper does not read well and is difficult to understand, as the theoretical discussion is separate from the simulation results and does not link back sufficiently.
Author Response

(The authors gave the same response as above.)

Round 2
Reviewer 2 Report
The author have answered my comments and I feel that the cahgnes to the layout have improved the manuscript sufficiently.